# Arsenic Oxidation and Removal from Water via Core–Shell MnO_2_@La(OH)_3_ Nanocomposite Adsorption

**DOI:** 10.3390/ijerph191710649

**Published:** 2022-08-26

**Authors:** Yulong Wang, Chen Guo, Lin Zhang, Xihao Lu, Yanhong Liu, Xuhui Li, Yangyang Wang, Shaofeng Wang

**Affiliations:** 1National Demonstration Center for Environmental and Planning, College of Geography and Environmental Science, Henan University, Kaifeng 475004, China; 2Henan Engineering Research Center for Control and Remediation of Soil Heavy Metal Pollution, Henan University, Kaifeng 475004, China; 3College of Software, Henan University, Kaifeng 475004, China; 4Key Laboratory of Industrial Ecology and Environmental Engineering (Ministry of Education, China), School of Environmental Science and Technology, Dalian University of Technology, Dalian 116024, China

**Keywords:** MnO_2_@La(OH)_3_ nanocomposite, arsenate, arsenite, removal, mechanism, oxidation

## Abstract

Arsenic (As(III)), more toxic and with less affinity than arsenate (As(V)), is hard to remove from the aqueous phase due to the lack of efficient adsorbents. In this study, a core–shell structured MnO_2_@La(OH)_3_ nanocomposite was synthesized via a facile two-step precipitation method. Its removal performance and mechanisms for As(V) and As(III) were investigated through batch adsorption experiments and a series of analysis methods including the transformation kinetics of arsenic species in As(III) removal, FTIR, XRD and XPS. Solution pH could significantly influence the removal efficiencies of arsenic. The adsorption process of As(V) occurred rapidly in the first 5 h and then gradually decreased, whereas the As(III) removal rate was relatively slower. The maximum adsorption capacities of As(V) and As(III) were up to 138.9 and 139.9 mg/g at pH 4.0, respectively. For As(V) removal, the inner-sphere complexes of lanthanum arsenate were formed through the ligand exchange reactions and coprecipitation. The oxidation of As(III) to the less toxic As(V) by δ-MnO_2_ and subsequently the synergistic adsorption process by the lanthanum hydroxide on the MnO_2_@La(OH)_3_ nanocomposite to form lanthanum arsenate were the dominant mechanisms of As(III) removal. XPS analysis indicated that approximately 20.6% of Mn in the nanocomposite after As(III) removal were Mn(II). Furthermore, a small amount of Mn(II) and La(III) were released into solution during the process of As(III) removal. These results confirm its efficient performance in the arsenic-containing water treatment, such as As(III)-contaminated groundwater used for irrigation and As(V)-contaminated industrial wastewater.

## 1. Introduction

The naturally ubiquitous occurrence of arsenic in surface waters and groundwater causes a serious threat to the ecosystem security and human health through its elevated levels of toxicity and accumulation [1]. The long-term ingestion of low concentrations of arsenic via drinking water and food in diet could result in chronic arseniasis, such as pigmentation, skin keratosis, disorders of vascular and nervous systems and cancers of skin and viscus [2]. As a consequence, the World Health Organization (WHO) proposed that the standard arsenic concentration in drinking water has been set to 10 μg/L in 1993 [3]. It has been reported that nearly 200 million people have been exposed to elevated arsenic concentrations in groundwater over the world [4]. In China, this risk endangers the safety of approximately 19.6 million people [5]. Hence, it is both urgent and essential to develop efficient technologies for removing arsenic from As-containing water.

Up to now, various approaches, including ion-exchange, adsorption, coagulation-precipitation, biological treatment and membrane filtration have been explored to decontaminate arsenic from aqueous systems [6,7,8,9,10,11,12,13,14,15]. Among these methods, adsorption has been proved to be a promising technique considering its low cost, convenient operation and high removal efficiency [16,17]. Correspondingly, great efforts have been devoted to preparing different adsorbents for drinking water purification, e.g., iron hydroxide and oxide, manganese oxides and polyaluminum [12,18,19,20,21,22]. However, many of these adsorbents suffer more or less challenges, such as incomplete removal, difficulty in the process of separation after the adsorption, inflated cost and the agglomeration of adsorbents [23,24]. Furthermore, arsenite (As(III)) exhibits more toxic and less strongly adsorbed to various adsorbents than arsenate (As(V)) [17,21]. Hence, much more explorations, such as the modification of biochar, should be made continuously to prepare facile adsorbents with a superior arsenic removal property, easier operation and lower treatment cost [25,26]. For instance, Benis et al. synthesized a binary oxide–biochar composite through a microwave pyrolysis combined with electrochemical modification in 2022 [27]. However, this composite biochar has some disadvantages: non-ideal adsorption capacity, complicated synthesis process and high cost, which are necessary to consider in the real implication.

Recently, La (lanthanum)-based materials applied for the treatment of arsenic-contaminated water environments have drawn particular attention for the extreme affinity of arsenic, a large number of adsorption sites, improved removal efficiencies and the excellent stability of adsorbed products [16,28,29,30]. The removal mechanisms of arsenic have been investigated through the studies of X-ray diffraction (XRD), Fourier transform infrared spectroscopy (FTIR), X-ray photoelectron spectroscopy (XPS), density function theory calculations (DFT) and extended X-ray adsorption fine structure (EXAFS) [16,28]. However, these La-based materials exhibited a limited removal ability of arsenite (As(III)), which is much more soluble, toxic and mobile than arsenate (As(V)) [21,31,32,33]. For example, the adsorption capacity of As(V) by La-impregnated activated alumina was 26.3 mg/g, whereas that of As(III) was 9.23 mg/g [16]. What is more, mesoporous graphene oxide–lanthanum fluoride nanocomposite, La-loaded orange waste gel and Fe-La composite (hydr)oxides reported in the previous literature were prepared for the removal of As(V), while the uptake performances of As(III) by these La-based materials have not been further studied, which could be due to the lower removal amount of As(III) [28,29,34]. Our previous research also suggested that the As(V) adsorption performance onto lanthanum hydroxide adsorbent was extremely excellent, while the adsorption amount of As(III) was particularly low [30]. This significant discrepancy could be attributed to the weak interaction between As(III) and La [2,17,21]. Therefore, the oxidation of As(III) to the less toxic As(V) in order to enhance the immobilization of arsenic and then removal via adsorption is an effective approach of As(III) removal [21,31,32,35,36,37].

Manganese oxide has been demonstrated to be effective in the oxidation of As(III) to As(V). Thus, many manganese oxide-based adsorbents have been designed and applied in the decontamination of both As(III) and As(V) [18,21,33,38,39,40,41]. Nevertheless, to the best of our knowledge, few studies have been conducted to synthesize Mn–La binary oxide adsorbents used for As(V) and As(III) removal. Su et al. proposed a manganese-doped lanthanum oxycarbonate adsorbent to decontaminate As(V) but the synthetic process was complicated, and the cost of preparation was expensive because of the use of high-temperature calcination and the removal property of As(III) was studied [39]. In order to dislodge As(III), the manganese-doped lanthanum oxycarbonate adsorbent was investigated to uptake arsenic through the process of the H_2_O_2_ oxidation and adsorption [36]. The difficulty of obtaining H_2_O_2_ and the safety of preservation and transport in the general natural environments are the major disadvantages of the pre-oxidation by the H_2_O_2_.

The objectives of the present study were to: (1) overcome the limitation for the low capacity of As(III) removal by La-based materials and improve the utilization efficiency of La in As(III) removal, (2) simultaneously oxidize and remove As(III) without the use of H_2_O_2_ and (3) prepare a MnO_2_ and lanthanum hydroxide composite material for As(V) and As(III) removal from water. Therefore, a novel core–shell structured δ-MnO_2_-doped lanthanum hydroxide (denoted as MnO_2_@La(OH)_3_) nanocomposite was prepared via a two-step precipitation method and applied as an adsorbent for the treatment of both As(V) and As(III) from aqueous solution. The structural characteristics of the synthetic MnO_2_@La(OH)_3_ were observed by field emission scanning electron microscopy with energy-dispersive X-ray spectroscopy (SEM-EDS), transmission electron microscopy (TEM) and high-resolution transmission electron microscopy (HRTEM). Batch adsorption experiments, including the effect of dosage and pH, adsorption isotherms, adsorption kinetics, the inhibiting influences of coexisting anions and the reusability, were evaluated to investigate the removal performance of As(V) and As(III). The special focus was on the changes of different arsenic species and the releases of La^3+^ and Mn^2+^ ions in solution during the removal process of As(III) in order to explore the oxidation–adsorption mechanisms of As(III) removal. The analyses of FTIR, XRD and XPS were further employed to evaluate the underlying mechanisms of As(V) and As(III) removal.

## 2. Materials and Methods

### 2.1. Materials

Chemical reagents including KMnO_4_, La(NO_3_)_3_∙6H_2_O, NaOH, As_2_O_5_ and As_2_O_3_, were of analytical grade and purchased from Sigma-Aldrich (Shanghai, China) without further purification. The stock solutions were all prepared by dissolving the required amount of reagents into deionized (DI) water, which was used for all experiments.

### 2.2. Preparation of MnO_2_@La(OH)_3_ Nanocomposite

The core–shell MnO_2_@La(OH)_3_ nanocomposite is synthesized by a two-step precipitation method as displayed in Figure 1. The δ-MnO_2_ (Layered, Birnessite) was prepared according to a previously described method [18,40]. Briefly, 16.8 g of KOH was dissolved in 100 mL of aqueous solution containing 46 mL of ethanol. An amount of 4.74 g of KMnO_4_ was dissolved in 150 mL DI water. Afterwards, the KOH solution was added slowly into the KMnO_4_ solution under violent stirring at room temperature. The obtained brown precipitate of δ-MnO_2_ was continuously stirred for 1 h. Then, a 50 mL solution containing 6.5 g of La(NO_3_)_3_∙6H_2_O was added dropwise to the abovementioned brown suspension. The resulting mixture was stirred in succession for 1 h and aged at room temperature for 3 h. Finally, the product of MnO_2_@La(OH)_3_ nanocomposite was centrifuged and washed three times with DI water, and then freeze-dried and stored in a desiccator for use in further tests.

### 2.3. Characterization of Adsorbents before and after as Removal

SEM-EDS (Hitachi S-4800, Tokyo, Japan) was employed to observe the morphologies and surface elemental compositions of the MnO_2_@La(OH)_3_ nanocomposite before and after arsenic removal. TEM and HRTEM were recorded on a G2 F20 S-TWIN (FEI, OSU, USA). XRD patterns were analysed using a D8 Advance X-ray diffractometer (Bruker-AXS, Karlsruhe, Germany). Brunauer–Emmett–Teller (BET) surface areas and pore structures of the material were measured via a surface area analyser (Micromeritics ASAP 2460, Portland, GE, USA) at 77 K using the N_2_ adsorption/desorption method. FTIR spectra were collected on a Fourier transform infrared spectrometer with a transmission model (Thermo Nicolet 6700, Waltham, MA, USA) using the KBr pellet method with 32 cumulative scans, the mid-infrared range of 400–4000 cm^−1^ and the spectral resolution of 4 cm^−1^. The XPS measurements were inspected on an ESCALAB 250Xi spectrometer (Thermo Scientific, Waltham, MA, USA) to analyse the surface composition and chemical states of elements in the samples. The zeta potential of the material was obtained by a Zetasizer Nano ZS (Nano 2000, Malvern Co., Malvern, UK) to estimate the pH of the point of zero charge (pH_pzc_) of the MnO_2_@La(OH)_3_ nanocomposite. In brief, the pH of the adsorbent of 0.5 g/L in the range of 2–11 was maintained constant for 3 h by the addition of NaOH or HCl solution, respectively, and the zeta potential was reported by the Zetasizer Nano ZS. The pH_pzc_ was determined at the point at which the value of the zeta potential was 0 in the curve of zeta potential as a function of pH.

### 2.4. Batch Adsorption Experiments

Batch experiments were performed by adding the MnO_2_@La(OH)_3_ nanocomposite into As solution at room temperature and a dosage of 0.5 g/L for 24 h unless otherwise stated. The effects of pH, adsorbent dosage and competitive anions (Cl^−^, SO_4_^2−^, NO_3_^−^, CH_3_COO^−^ (Ac^−^), PO_4_^3−^ and citrate ions) on the uptake of As(V) and As(III) were conducted at initial As concentrations of 65 mg/L. The adsorption kinetics studies were examined at pH 4.0 for 30 h with the initial As concentrations of 65 mg/L and the supernatant samples were collected via filtration at certain intervals. The adsorption isotherms were measured at pH 4.0 and the initial concentrations of arsenic in solution ranged from 40.0 to 100.0 mg∙L^−1^. Adsorption–desorption experiments were studied four consecutive times by the addition of 0.5 g/L of the adsorbent to 65 mg/L of arsenic solution at pH 4.0 and a 2.0 M NaOH solution was applied to regenerate over 4 h. The pH of the solution was controlled at predetermined pH values by adding 1 M NaOH or HCl solution throughout the removal processes. After the adsorption experiments, the suspensions were filtrated by a 0.22 μm filter and the residual arsenic concentration in the solution was determined by a hydride-generation atomic fluorescence spectrophotometer (HG-AFS, AFS-3100, Haiguang Corp., Beijing, China).

To further understand the oxidation and adsorption process of As(III), transformations of arsenic species were performed in a similar experimental process as adsorption kinetics test except for the initial As concentration of 75 mg/L. The concentrations of lanthanum and manganese ions in solution were determined by an inductively coupled plasma emission spectrometry (ICP-OES, iCAP-6000, Waltham, MA, USA). The content of the total amount of As (As(T)) and As(III) in solution was detected by the HG-AFS according to the literature previously described [42,43]. The concentration of As(T) was quantified by a method of reduction with 5% thiourea/ascorbic acid and acidification with 5% HCl solution before the hydride generation, whereas a technique of disodium citrate buffer (0.5 M) was applied for the As(III) detection. The difference between As(T) and As(III) was the As(V) concentration. All batch experiments were examined in a thermostatic shaker at 180 rpm in triplicate and the average values of the results were reported.

### 2.5. Statistical Analysis

The kinetic data were evaluated by the pseudo-first order and pseudo-second order models. The linear forms of these two models are calculated as follows [44,45]:

The pseudo-first-order model:(1)$$\ln(q_{e}-q_{t})=\ln q_{e}-k_{1}t$$

The pseudo-second-order model:(2)$$\frac{t}{q_{t}}=\frac{1}{k_{2}q_{e}{}^{2}}+\frac{t}{q_{e}}$$
where *q*_e_ and *q*_t_ represent the adsorption amounts (mg/g) of arsenic at equilibrium and at time *t* (h), respectively, and *k*_1_ (h^−1^) and *k*_2_ (g/(mg∙h)) are the rate constants of the two models, respectively.

The adsorption isotherms were studied by the Freundlich and Langmuir models. The linear expressions of these isotherm models are listed as follows [8,44]:(3)$$\frac{C_{e}}{q_{e}}=\frac{1}{q_{\max}}C_{e}+\frac{1}{K_{L}q_{\max}}$$
(4)$$\log q_{e}=\log K_{F}+\frac{1}{n}\log C_{e}$$
where *C*_e_ (mg/L) represents the arsenic concentration at equilibrium; *q*_e_ and *q*_max_ are the equilibrium and maximum adsorption capacities (mg/g), respectively; *K*_L_ (L/mg) and *K*_F_ ((mg/g)·(mg/L)^–1/*n*^) are the adsorption constants for the Langmuir and Freundlich models, respectively; and 1/*n* represents a heterogeneity factor related to adsorption intensity. *K*_L_ and *K*_F_ roughly stand for the adsorption affinity between the active sites and arsenic.

The reference parameters were estimated via nonlinear regression by the Statistic 8.0 software (Statsoft, EUA, Tulsa, OK, USA). The coefficient of determination (*R*^2^), mean square error (MSE) and average relative error (ARE) were calculated to evaluate the goodness of fit [46,47,48]. The data were subjected to analysis of variance (ANOVA) using the General Linear Model and significance was declared with *p* < 0.05. The post hoc analysis was carried out using the Tukey’s test.

## 3. Results and Discussion

### 3.1. Characterization of MnO_2_@La(OH)_3_ Nanocomposite

#### 3.1.1. SEM and TEM Analysis

The particle surface morphology and core–shell structure of the MnO_2_@La(OH)_3_ nanocomposite were performed by SEM-EDS and TEM. As shown in Figure 2a, the SEM image of the virgin material suggests that the surface of the particles was rough and had been formed by the aggregates of the uneven quasi-spherical nanoparticles with a diameter of 5–100 nm. The aggregation of the nanoparticles resulted into the formation of heterogeneous and irregular pore channels with various sizes. The EDS analysis of the original adsorbent shows that the elements of lanthanum, manganese and oxygen were detected on the surface. The molar ratio of La and Mn atoms was approximately 6.55, which was much higher than that (1:2) used in the preparation process. This could be attributed to the limited detection of most of the Mn atoms inside the particles, indicating the successful synthesis of the core–shell structure of the MnO_2_@La(OH)_3_ nanocomposite. After the As(V) removal, nanorod structures could be observed (Figure 2b), suggesting that a new compound was formed. The new compound was further determined as lanthanum arsenate by the XRD analysis as presented in Figure 6c. However, the surface of the irregular nanoparticles became smoother after the uptake of As(III) (Figure 2c), which could be due to the occurrence of the redox reaction between As(III) and MnO_2_ and the precipitation of manganese hydroxide on the surface of the subparticles. Through the EDX analysis, La, Mn, O and As elements were observed on the surface after the uptake of As(V) and As(III), and the presence of As element suggested the successful immobilization of As by the MnO_2_@La(OH)_3_ nanocomposite. What is more, the La/Mn molar ratio after arsenic removal was close to 1:2, which was in line with that used in the synthesis. This could be attributed to the release of La on the surface into solution under acidic condition. The EDX analysis of the MnO_2_@La(OH)_3_ nanocomposite before and after the removal of As(V) and As(III) combined the synthesis process of the material indicated the structure of the synthetic MnO_2_@La(OH)_3_ nanocomposite was core–shell.

Figure 3a,b illustrates the high-magnification TEM image and HR-TEM image of the MnO_2_@La(OH)_3_ nanocomposite. As depicted in Figure 3a, it is remarkable that the MnO_2_@La(OH)_3_ nanocomposite was accumulated by the stratiform of the nanoparticles. The HR-TEM image demonstrates that the lattice fringes at the edges of nanoparticles separated by *d* = 0.23, 0.32 and 0.33 nm could be assigned to the (2 0 1), (1 0 1) and (1 1 0) planes of the hexagonal lanthanum hydroxide (Figure 3b). Nevertheless, no clear lattice fringes could be found in the center, which could be due to the presence of MnO_2_ with poor crystallinity. This observation further suggested that the structure of the MnO_2_@La(OH)_3_ nanocomposite was yolk–shell.

#### 3.1.2. Specific Surface Area

As shown in Figure 3c, it can be observed that according to the IUPAC classification, the adsorbent exhibited a type IV isotherm with H3-type hysteresis loops for its isotherms did not exhibit a plateau at high p/p_0_ values. This result indicated that multilayered physical adsorption occurred on the aggregates of platelet-like particles and that the nanocomposite was mesoporous, as illustrated in the inset in Figure 3c [8,49]. Furthermore, the specific surface area estimated using the BET method was 37.90 m^2^/g. Based on the analysis of the BJH model using the desorption branch, the average pore diameter was 9.56 nm and the total pore volume was 0.16 cm^3^/g. No microporosity by the t-plot method had been observed in the material. The mesoporous pore size distribution of the MnO_2_@La(OH)_3_ nanocomposite is considered to be appropriate for the uptake of arsenic.

#### 3.1.3. Point of Zero Charge

Furthermore, the pH_pzc_ of the MnO_2_@La(OH)_3_ nanocomposite was estimated by the method of surface zeta potential as a function of solution pH to underlie the effect of solution pH on arsenic removal. As demonstrated in Figure 3d, it can be clearly observed that the surface charge of the material significantly depended on the solution pH, and the pH_pzc_ value was determined to be approximately 2.7. According to the previous literature, the pH_pzc_ values of δ-MnO_2_ and lanthanum hydroxide were approximately 1.3 and 9.5, respectively [30,40,50,51,52]. The composite of δ-MnO_2_ and lanthanum hydroxide could be the reason, which resulted into the pH_pzc_ value of the material between 1.3 and 9.5. The release of La due to the dissolution under the acidic condition (Figure 4a,b) might cause the exposure of the MnO_2_ in shell. This resulted into that the pH_pzc_ value was closed to the pH_pzc_ value of δ-MnO_2_ (1.3), which shifted to the lower pH value compared with that of the pure lanthanum hydroxide [30,51,52,53]. What is more, the pH_pzc_ value of a Mn–La composite reported by Yu and Chen was about 6.2 [45]. This discrepancy could be attributed to the manganese (hydr)oxide in the Mn–La composite. Guo et al. suggested that the pH_pzc_ value of MnOOH nanorods was determined at approximately 4.3 [54]. Previous studies suggested that the surface of the materials was positive due to the protonation at pH < pH_pzc_, resulting in the enhancement of electrostatic attraction between arsenic and the active sites of the materials [23,24,45,49,55,56]. This would be beneficial for the arsenic removal. However, as the solution pH was above its pH_pzc_, the formation of the negatively charged surface could result in a stronger electrostatic repulsion between arsenic and the adsorption sites, which would be adverse for arsenic adsorption and lead to the decrease of arsenic adsorption capacities.

### 3.2. Arsenic Adsorption Performance

#### 3.2.1. Adsorption Kinetics

In order to evaluate the adsorption rate of As(V) and As(III) onto the MnO_2_@La(OH)_3_ nanocomposite, the amount of arsenic adsorption depending on contact time was investigated at pH 4.0 with an initial As concentration of 65 mg/L. As illustrated in Figure 4a, it can be observed that the initial As(V) adsorption process was very fast in the first 5 h, and then gradually slowed down. The adsorption equilibrium was achieved in 24 h. Sufficient active sites with strong affinity for As(V) species on the surface of the MnO_2_@La(OH)_3_ nanocomposite were the main cause of the rapid adsorption process at the preliminary stage. With the decrease of the active sites, the adsorption rate gradually decreased and intraparticle diffusion and surface precipitation controlled the uptake of arsenic [12,20,57]. Similar processes of As(V) adsorption have also been reported in the As(V) adsorption behaviours by other types of adsorbents [12,23,56,57]. However, compared with the As(V) adsorption process, the As(III) removal was a relatively slower process and could reach equilibrium after 24 h (Figure 4b). This could be due to the necessity of the oxidation of As(III) to As(V) by δ-MnO_2_ before the uptake by lanthanum hydroxide. The content of δ-MnO_2_ in the MnO_2_@La(OH)_3_ nanocomposite might determine the oxidation rate of As(III). The transformations of arsenic species in solution as a function of reaction time were further investigated as shown in Figure 6a.

To better understand the adsorption processes of As(V) and As(III), the experimental data obtained from adsorption kinetics were simulated by two empirical adsorption reaction models of the pseudo-first-order model and the pseudo-second-order models [44,49]. The fitting plots of the kinetic models are displayed in Figure 4a,b, and the acquired parameters and correlation coefficients are shown in Table 1. It is manifest that the adsorption process of As(V) was described better by the pseudo-second-order model with a higher value of correlation coefficient (*R*^2^ > 0.99) and lower values of MSE and ARE than the pseudo-first-order model. This result indicated that the main adsorption process of As(V) by the MnO_2_@La(OH)_3_ nanocomposite involved chemisorption process [8,44]. The pseudo-second-order model supposed that the adsorption capacities of these materials were predominantly proportional to the square of the number of unoccupied sites on the surface [45]. The As(V) adsorption capacity of *q*_e_ and adsorption rate constant of *k*_2_ estimated by the pseudo-second-order model were 129.87 mg/g and 0.00872 g/(mg∙h), respectively. However, for the As(III) adsorption kinetics, the pseudo-first-order model had a higher value of correlation coefficients (*R*^2^ > 0.99) and lower values of MSE and ARE, indicating that the kinetics data of As(III) adsorption can be fitted better by the pseudo-first-order model compared with the pseudo-second-order model. The discrepancy between the As(III) adsorption kinetics and the As(V) adsorption kinetics could be explained by the process of the As(III) oxidation and subsequent adsorption. Moreover, the As(III) adsorption capacity of *q*_e_ and adsorption rate constant of *k*_1_ estimated by the pseudo-first-order model were 140.54 mg/g and 0.098 h^−1^, respectively. The As(III) adsorption capacity of *q*_e_ acquired by the pseudo-second-order model was 127.8 mg/g, which was closer to the experimental adsorption capacity (127.9 mg/g). The maximum adsorption capacity of arsenic by the MnO_2_@La(OH)_3_ nanocomposite at equilibrium might be controlled by the content of lanthanum hydroxide in the nanocomposite.

#### 3.2.2. Effect of Adsorbent Dosage

As presented in Appendix A, the effect of MnO_2_@La(OH)_3_ dosage on arsenic removal was performed by varying the adsorbent concentrations at pH 4.0. Obviously, the removal efficiencies of both As(V) and As(III) significantly increased with the increase of the adsorbent dosage and levelled off as the dosage was above 0.5 g/L. The efficiencies of As(V) and As(III) reached 96.0% and 97.4% at a dose of 0.5 g/L, respectively. Specifically, when the dose was below 0.4 g/L, the efficiency of As(V) removal was a little higher than that of As(III) removal at the same adsorbent dosage. This comparison was different from the effect of solution pH (Appendix A), which could be attributed to the La release at pH 4.0. Therefore, the dose of 0.5 g/L was considered to be optimized in order to further investigate the batch adsorption experiments.

#### 3.2.3. Effect of Solution pH

Solution pH is one of the most important influence factors of arsenic removal for both the surface zeta potential of adsorbents (Figure 3d) and arsenic species in solution can be significantly affected by the solution pH. Consequently, the effect of the solution pH on the removal efficiency of As(V) and As(III) was carried out in a solution with the initial As concentration of 65 mg/L at pH values ranging from 3–11, and the results for As(V) and As(III) are illustrated in Figure 4c,d, respectively. It can be obviously observed that the removal efficiencies of both As(V) and As(III) increased from pH 3 to 4 and decreased remarkably from pH 4 to 11. The optimal arsenic removal efficiencies were observed with 95.9% for As(V) and 97.7% for As(III) at pH 4.0. The trends for As(V) and As(III) removal affected by the solution pH were similar. Similar trends for As(V) and As(III) removal by other oxidizing adsorbents, such as Fe–Mn binary oxide adsorbent have also been observed [40,41,55]. The homologous observation between As(V) and As(III) could be attributed to the process of the oxidation and adsorption of As(III) to As(V) [31,55].

The adsorption behaviour of As(V) by the MnO_2_@La(OH)_3_ nanocomposite at pH > 4.0 can be interpreted by the distribution of arsenate species, the surface charge of the adsorbent in solution and the competition between arsenate species and hydroxyl ions in solution [31,44,49]. As displayed in Appendix A, the predominant species in solution were H_2_AsO_4_^−^ and HAsO_4_^2−^ and the surface of the material was negatively charged. The enhancement of electrostatic repulsion between the surface adsorption sites of the material and arsenate species significantly decreased the arsenic removal efficiency. Studies indicated that the La(III) ion could be released into the solution under acidic condition (pH < 5.0 in this study) [28,45,58,59,60,61]. Up to 23.6% of La in the La(OH)_3_/Fe_3_O_4_ adsorbent dissolved at pH value of 3.6 [58]. Therefore, the release of La dissolving into solution resulted in the decrease of arsenic removal efficiency from pH 4.0 to 3.0.

Different from the As(V) removal, the oxidation of As(III) to As(V) by δ-MnO_2_ and then removal by lanthanum hydroxide could be the main removal mechanisms of As(III) by the MnO_2_@La(OH)_3_ nanocomposite. It should be noted that our previous study indicated that lanthanum hydroxide exhibited excellent removal properties for As(V), whereas negligible removal capacities for As(III) were extant [30]. As depicted in Figure 4c,d, La(III) and Mn(II) ions release in solution from the MnO_2_@La(OH)_3_ nanocomposite after As(V) and As(III) removal at different pH conditions were also detected. For the removal of As(V), the La release was 35.4 mg/L at pH 3.0, and reduced with the increment of pH values, whereas the release of La could be ignored at pH values > 5.0. The release of La as a function of pH have also been reported by other La-based adsorbents in previous literature, such as lanthanum hydroxide materials, magnetic porous biochar supported La(OH)_3_, La(OH)_3_/Fe_3_O_4_ nanocomposites and lanthanum hydroxide-doped activated carbon fiber [58,59,60,61]. However, the Mn(II) release can be negligible within all pH experimental conditions. For the As(III) removal, a similar trend of La release to that of As(V) removal could be observed, while the Mn(II) release decreased from 32.3 mg/L at pH 3.0 to undetectable concentrations at pH values higher than 8.0. Similar Mn(II) release during the process of As(III) oxidation has also been described in previous studies [18,31,38,40,41]. The release kinetics of La(III) and Mn(II) ions during the process of As(III) removal by the MnO_2_@La(OH)_3_ nanocomposite at pH 4.0 would be further discussed as below (Figure 6a in Section 3.3).

Furthermore, as shown in Appendix A, the comparison of As(V) and As(III) removal with the effects of solution pH indicated that the adsorption amounts of As(III) by the MnO_2_@La(OH)_3_ nanocomposite were slightly higher than those of As(V) adsorption at the same pH value. This observation could be attributed to the formation of manganese (hydr)oxides by the precipitation of Mn(II) [54]. It should be noted that an As(III) adsorption capacity by hydrous manganite (MnOOH) nanorods at pH 7.0 was found to be over 431.2 mg/g [54].

#### 3.2.4. Adsorption Isotherms

To estimate the maximum adsorption capacities of As(V) and As(III) on the MnO_2_@La(OH)_3_ nanocomposite, arsenic adsorption isotherms were performed at pH 4.0 with various initial As concentrations and two isotherm models, including Langmuir and Freundlich models, were used to simulate the adsorption experimental data [44,45,49]. The experimental results and fitting curves are depicted in Figure 4e,f, and the corresponding isotherm parameters are illustrated in Table 2. As shown in Table 2, it can be found that both the Langmuir and Freundlich models had high regression coefficients (*R*^2^ > 0.99), suggesting that both models could simulate the adsorption data well. However, the lower values of MSE and ARE indicated that the adsorption isotherms data could be represented better by the Freundlich model than the Langmuir model. The Langmuir model theory considers that the adsorption is approximately monolayer adsorption and that the adsorbed sites are homogeneously covered on the surface of the adsorbents [44]. On the other hand, the Freundlich model assumed that the adsorption occurred through the multilayer and heterogeneous surfaces of the adsorbents [27]. In addition, the maximum adsorption capacities for As(V) and As(III) calculated by the Langmuir model were 138.9 and 139.9 mg/g, respectively. These results indicated that the δ-MnO_2_ and lanthanum hydroxide in the nanocomposite were uniform and the structure was beneficial for the removal of As(V) and As(III). These observations also suggested that most As(III) species in solution were oxidized to As(V) by δ-MnO_2_ and then removed by lanthanum hydroxide in the material due to the negligible adsorption amount of As(III) by lanthanum hydroxide and the lower adsorption capacity by δ-MnO_2_ [18,30,38]. These results were further confirmed by the XRD and XPS analyses as below (Figures 6c and 7).

Furthermore, the maximum capacities of arsenic adsorption on different adsorbents were compared in Table 3. The superior removal performances of As(V) and As(III) by the MnO_2_@La(OH)_3_ nanocomposite were significantly higher than those of most of currently reported adsorbents, indicating an efficient potential application for the decontamination of As-containing water, especially containing As(III).

#### 3.2.5. Effect of Co-Existing Oxyanions

Arsenic in natural water and industrial wastewater generally coexist with various anions, such as Cl^−^, SO_4_^2−^, NO_3_^−^, PO_4_^3−^ and organic acids. The presence of these inorganic anions could compete with arsenic for the available active sites on the material, which may influence the removal efficiency of arsenic [55]. Organic acids in solution can react with La^3+^ ion to form La-coordination complexes, resulting in the release of La from the adsorbent and the decrease of the arsenic removal efficiencies [49]. Therefore, the influences of inorganic anions (Cl^−^, SO_4_^2−^, NO_3_^−^ and PO_4_^3−^) and organic acids (Ac^−^ and citrate) on the removal efficiency of As(V) and As(III) by the MnO_2_@La(OH)_3_ nanocomposite were performed at pH 4.0 with the initial As concentration of 65.0 mg/L along with two different concentrations of co-existing anions. As presented in Figure 5a,b, it can be obviously seen that the effects of co-existing anions on the removal efficiency of As(V) and As(III) were similar to each other, indicated by the statistical analysis using the ANOVA with Tukey’s test. These observations were consistent with the results of the adsorption envelope study (Appendix A). The co-existing Cl^−^ and NO_3_^−^ showed a slight effect on removal efficiencies. Less than 5% of removal efficiency was reduced even when the concentrations of Cl^−^ and NO_3_^−^ were up to 50 mM, whereas SO_4_^2^^−^ and PO_4_^3^^−^ exhibited significant inhibitory effects on the arsenic removal. Remarkably, the decreases in the removal efficiency of As(V) and As(III) due to the presence of PO_4_^3^^−^ were more than the decreases at the presence of SO_4_^2^^−^. The most significant reduction of arsenic removal in the presence of phosphate could be attributed to the effective competition between arsenic species and phosphate for their similarities in coordination geometry and geochemical behaviours [13,66]. The competitions between arsenic species and Ac^−^ were slight under the lower concentration (10 mM) of Ac^−^, while the interferences were drastic under the higher concentration (50 mM). It should be noted that 2 mM of citrate could lead to the complete dissolution of the MnO_2_@La(OH)_3_ nanocomposite. Therefore, the effects of citrate on As removal were not presented in Figure 5. These results indicated that the co-existing citrate manifested the most significant decrease of arsenic uptake due to the release of La coordinated with citrate under acidic condition. In general, the inhibiting effects of these anions on As removal followed the order: citrate > PO_4_^3−^ > SO_4_^2−^ > Ac^−^ > NO_3_^−^ ≈ Cl^−^. Similar observations have also been found in previous studies [8,28,31,33,55,64].

#### 3.2.6. Regeneration and Reusability

Considering the potential application and economic cost, the consecutive adsorption-desorption experiment was investigated at pH 4.0 to perform the reusability of MnO_2_@La(OH)_3_ nanocomposite. As given in Figure 5c, the adsorption capacities of As(V) and As(III) after the first four regenerations decreased substantially from 123.1 and 123.9 mg/g for the virgin adsorbent to 47.6 and 55.6 mg/g, respectively. The significant reduction could be due to the formation of lanthanum arsenate after arsenic removal, which was the irreversible occupancy of the available binding sites [28,30]. From the second to the fourth cycle, the removal efficiencies of As(V) and As(III) were reduced and then remained stabile. However, the adsorption capacities remained higher than 28.7 mg/g for As(V) and 25.8 mg/g for As(III). In view of that arsenic concentrations in natural water are usually less than 1 mg/L [4], this material has promising applications for treating the As-contaminated water, particularly the As(III)-containing groundwater.

### 3.3. Adsorption Mechanisms

Based on the above experimental results on the adsorption of As(V) and As(III), a series of analysis methods were applied to characterize the MnO_2_@La(OH)_3_ nanocomposite before and after arsenic removal in order to further explore the removal mechanisms of As(V) and As(III).

To further verify the oxidation process of As(III) to As(V) during the As(III) removal by the MnO_2_@La(OH)_3_ nanocomposite, the transformation kinetics of arsenic species in solution were evaluated at pH 4.0 with the initial As(III) concentration of 75 mg/L. As demonstrated in Figure 6a, it can be seen that the concentrations of As(III) and As(T) in solution decreased sharply at the first 15 h, and then turned to a relatively slower rate. These observations were consistent with the As(III) adsorption kinetics (Figure 4d). Almost 92% of the As(III) was removed at the equilibrium. Due to the oxidation of As(III) to As(V) by δ-MnO_2_, the concentration of As(V) in solution rose up to approximately 7.6 mg/L at the first 2 h and decreased gradually to approximately 2.6 mg/L. The adsorption and immobilization by the lanthanum hydroxide on the nanocomposite resulted in the reduction of As(V) in solution. This result indicated that the transformation of As(III) to As(V) as oxidized by δ-MnO_2_ was rapid and that the oxidation and adsorption by the MnO_2_@La(OH)_3_ nanocomposite reacted simultaneously. The observation of the change of As(V) concentration in solution has been previously reported by the oxidizing adsorbents [31,40,41]. Furthermore, the Mn(II) in solution gradually increased to about 28.7 mg/L at the 22th hour with the progress of oxidation reaction and then slightly reduced to around 27.2 mg/L, which could be attributed to the formation of manganite hydroxide precipitation. As presented in Figure 6a, the release of La(III) due to the dissolution of lanthanum hydroxide and the formation of Mn(II) peaked at about 7.4 mg/L at 3.5 h. The decrease of the La(III) release could be attributed to the formation of the lanthanum arsenate coprecipitation, which was further confirmed by the XRD analysis.

The FTIR spectra of the MnO_2_@La(OH)_3_ nanocomposite before and after arsenic removal are shown in Figure 6b. Generally, the peaks at around 3432 and 1634 cm^−1^ could be attributed to the stretching vibration and bending vibration of the –OH groups and physisorbed water molecules [31,45,67]. The peaks centered at 1397, 864, 626 and 525 cm^−1^ are the characteristics of La-based bimetal hydroxides [28,45,53]. The band of pristine (oxy)hydroxides located at 1506 cm^−1^ could be assigned to the vibration of the hydroxyl groups bonding with the La atoms (La–OH bonds) [45]. After arsenic removal, two new vibration bands were observed at 850 and 450 cm^−1^. The peak located at 850 cm^−1^ could be ascribed to the stretching vibration of the As–O bonds, and the vibration band observed at 450 cm^−1^ could correspond to the bending vibration of the As–O groups [68]. These two peaks could be assigned to the characteristic vibration bands of lanthanum arsenate, which was further confirmed by XRD analysis (Figure 6c). I can be seen that after the uptake of arsenic, the intensity of the peak at approximately 1506 cm^−1^ almost disappeared, suggesting that the ligand exchange reactions of hydroxyl groups between the arsenic and La–OH groups were the main removal mechanism, and the inner-sphere surface complexes were formed.

As illustrated in Figure 6c, the X-ray diffraction patterns of the MnO_2_@La(OH)_3_ nanocomposite before and after arsenic adsorption at pH 4.0 were compared and indexed on the basis of the Joint Committee on Powder Diffraction Standards (JCPDS). The diffraction peaks of the MnO_2_@La(OH)_3_ nanocomposite can be well identified as the hexagonal phase of lanthanum hydroxide (La(OH)_3_, JCPDS 36-1481), while the typical peaks of δ-MnO_2_ were not able to be observed, which could be due to its poor crystallization [18,40]. After the As(V) and As(III) removal, the peaks attributed to the original material disappeared and the characteristic peaks were observed at 2*θ* values of 26.2°, 28.0°, 29.9° and 46.7°, which were well consistent with the monoclinic phase of lanthanum arsenate (LaAsO_4_, JCPDS 36-1481). This result sufficiently indicated that arsenic in solution was successfully retained by the adsorbent, as well as the lanthanum hydroxide in the MnO_2_@La(OH)_3_ nanocomposite transformed into lanthanum arsenate after the uptake of As(V) and As(III).

The XPS spectra were employed to gain insights into the surface elemental composition and the bonding configuration of MnO_2_@La(OH)_3_ nanocomposite before and after arsenic removal and the results are depicted in Figure 7. As displayed in Figure 7a, the full spectrum of the MnO_2_@La(OH)_3_ nanocomposite suggests the presence of Mn, La and O elements, and the appearance of As element peaking at ~45.0 eV after the As(III) and As(V) removal was in line with the uptake process of arsenic. The high-resolution XPS spectra of La 3d before and after arsenic removal are mainly composed of La 3d_5/2_ and La 3d_3/2_ due to the spin–orbit splitting, which is illustrated in Figure 7b. For the original material, the typical peaks of La 3d_5/2_ were centered at 834.5 and 838.5 eV, and the representative peaks of La 3d_3/2_ peaked at 851.3 and 855.3 eV. It is obvious that the binding energies of La 3d_5/2_ and La 3d_3/2_ after the arsenic removal shifted to higher values (approximately 0.4–0.8 eV), which suggested the possible electron transfer in the valence band of La 3d and the formation of the La–O–As inner-sphere complexation [30,31,49]. This observation has also been reported in the previous literature after phosphate removal [58,61,69]. Figure 7c exhibits the As 3d spectra of the three samples. After the As(III) and As(V) removal, the peaks are located at 45.11 and 45.13 eV. This result indicates that the arsenic immobilized in the MnO_2_@La(OH)_3_ nanocomposite after As(III) removal were almost As(V) species. Our previous research found that lanthanum hydroxide could selectively remove As(V), whereas the amount of As(III) removed by lanthanum hydroxide was quite low [30]. To further explore the arsenic species on the MnO_2_@La(OH)_3_ nanocomposite after the uptake of As(III), the As 3d spectrum was deconvoluted into two peaks at 44.1 and 45.1 eV, which corresponded to As(III) and As(V), respectively [33,55,70]. The result of the XPS analysis indicates that approximately 93.2% of As(III) were oxidized to As(V) by δ-MnO_2_ in the MnO_2_@La(OH)_3_ nanocomposite.

In order to prove the functionality of Mn(IV) oxidation and determine the valence state of Mn element, Figure 7d presents the high-resolution XPS spectra of Mn 2p. The Mn 2p spectrum of the virgin MnO_2_@La(OH)_3_ nanocomposite is composed of two peaks with the binding energy values at 642.7 and 654.2 eV, respectively, which could be assigned to Mn(IV) 2p_3/2_ and Mn(IV) 2p_1/2_, respectively [31,71]. After the As(V) removal, no evident divergence between the spectrum of the nanocomposite and that of the original material demonstrated that the valence state of Mn had not changed. Nevertheless, the form and intensity of the Mn 2p spectrum changed significantly after the removal of As(III). The Mn 2p spectrum peaks could be deconvoluted into three peaks, and a new peak appeared at the binding energy of 641.7 eV, which could be identified as the Mn(II) 2p_3/2_. As was seen through XPS analysis, about 20.6% of Mn(II) were present in the MnO_2_@La(OH)_3_ nanocomposite after As(III) removal, suggesting that chemical reaction between Mn(IV) and As(III) occurred. This phenomenon was in consist with the result of the As 3d XPS analysis (Figure 7c).

The high-resolution XPS scans of O 1s spectra of the MnO_2_@La(OH)_3_ nanocomposite before and after As(V) and As(III) are displayed in Figure 7e. The peak of the O 1s of the virgin adsorbent is centered at 530.78 eV. After As(III) and As(V) removal, the peaks of O 1s are located at 530.95 and 530.98 eV, respectively. The shift to higher binding energies after arsenic adsorption could be attributed to the formation of the La–O–As inner-sphere complexation [16,28,30]. As given in Figure 7e, the O1s spectrum of the MnO_2_@La(OH)_3_ nanocomposite could be fitted into three peaks at 529.77, 530.34 and 532.55 eV, corresponding to different oxidation forms including the lattice oxygen species of La–O and Mn–O (O_lat_), the surface chemisorbed oxygen species of hydroxyl groups bonded to the metal (O_sur_) and the adsorbed water on the surface (H_2_O), respectively [31,44,49]. After the uptake of As(III) and As(V), new component peaks attributed to arsenic–oxygen bonds (As–O) could be observed at 531.46 and 531.69 eV, respectively [33,55,72]. The relative concentrations of the peaks identified as the As–O bond were 26.3% and 25.3% for the As(III) and As(V) removal, respectively. In addition, after the removal of As(III) and As(V), the relative area ratios of O_sur_ groups decreased dramatically from 60.1% to 29.1% and 25.7%, respectively, indicating the formation of inner-sphere complexes through surface hydroxyl exchange reactions between the O_sur_ species on the adsorbent and the arsenic species [11,13,33,56,63]. This observation was in agreement with that proven by the abovementioned FTIR investigation discussed (Figure 6b). The exploration of removal mechanism has also been demonstrated in former studies on the uptake of phosphate via the XPS analysis [44,49,61].

Based on the above analyses and discussion, for the As(V) removal, the major removal mechanism is likely to be the formation of lanthanum arsenate by the ligand exchange reactions between the hydroxyl groups on the surface of the MnO_2_@La(OH)_3_ nanocomposite (O_sur_) and the arsenic species, described as follows:(5)$${\mathrm{Mn}}^{\mathrm{IV}}O_{2}@\mathrm{La}{\left(\mathrm{OH}\right)}_{3}(s)\;+{\;H}_{2}{\mathrm{AsO}}_{4}^{-}{+H}^{+}\to {\;\mathrm{Mn}}^{\mathrm{IV}}O_{2}@{\mathrm{LaAsO}}_{4}+{\;H}_{2}O$$

For the As(III) removal, most of the As(III) species were oxidized to As(V) species by δ-MnO_2_, a small amount of Mn^2+^ were released into the solution and then immobilized by the lanthanum hydroxide to form the lanthanum arsenate compound, expressed in the following:(6)$$\begin{array}{l} {\mathrm{Mn}}^{\mathrm{IV}}O_{2}@\mathrm{La}{\left(\mathrm{OH}\right)}_{3}(s)\;+{\;H}_{3}{\mathrm{As}}^{\mathrm{III}}O_{3}{+H}^{+}\to {\;\mathrm{Mn}}^{\mathrm{IV}}O_{2}@{\mathrm{LaAsO}}_{4}{\mathrm{Mn}}^{\mathrm{II}}{\left(\mathrm{OH}\right)}_{2}+\\ \;\;\;\;\;\;\;\;\;\;\;\;\;\;\;\;\;{\;\mathrm{Mn}}^{2+}(\mathrm{aq})+H_{2}O \end{array}$$

Moreover, the surface coprecipitation between La(III) and As(V) was another mechanism of the formation of lanthanum arsenate under acidic condition, represented as follows:(7)$${\mathrm{La}}^{3+}(\mathrm{aq})\;+{\;H}_{2}{\mathrm{AsO}}_{{}_{4}}^{-}\;\to {\;\mathrm{LaAsO}}_{4}$$

Overall, the MnO_2_@La(OH)_3_ nanocomposite showed efficient performance in removing As(III) from the aqueous solution, coupling the process of chemisorption and oxidation, compared with the non-ideal adsorption capacities of the La-based adsorbents in As(III) removal [21,31,32,33]. Its operation in As(III) removal was simple, owing to the non-utilization of the additional oxidants (such as H_2_O_2_) or visible-light irradiation [37,71]. Thus, it shows high potential for the treatment of high As-contaminated water.

## 4. Conclusions

In conclusion, a core–shell structured MnO_2_@La(OH)_3_ nanocomposite was fabricated and employed to remove As(III) and As(V) from water. The adsorption batch experiments suggested that the MnO_2_@La(OH)_3_ nanocomposite exhibited excellent performance in As(III) and As(V) removal. The As(V) adsorption process was very rapid in the first 5 h, and then gradually reduced, while the As(III) removal underwent a relatively slower process for the necessity of the As(III) oxidation. The solution pH and co-existing anions can significantly influence the arsenic removal efficiencies. Based on the results of the transformation kinetics in As(III) removal and the FTIR, XRD and XPS studies, the predominant mechanisms of As(V) removal were the ligand exchange reactions of surface hydroxyl groups (La–OH) and the surface coprecipitation between La(III) and As(V), which formed the inner-sphere surface complexes of lanthanum arsenate. The oxidation of As(III) to As(V) by δ-MnO_2_ and subsequently synergistic adsorption process by the lanthanum hydroxide on the MnO_2_@La(OH)_3_ nanocomposite to form lanthanum arsenate was the main removal mechanism of As(III). What is more, the XPS analysis indicated that approximately 20.6% of Mn(II) presented in the MnO_2_@La(OH)_3_ nanocomposite after As(III) removal. The prepared MnO_2_@La(OH)_3_ nanocomposite exhibited superior performance in the uptake of arsenic, especially in the simultaneous oxidation and removal of As(III), indicating an immense potential for the decontamination of As-containing groundwater.

## Figures and Tables

**Figure 1 ijerph-19-10649-f001:**
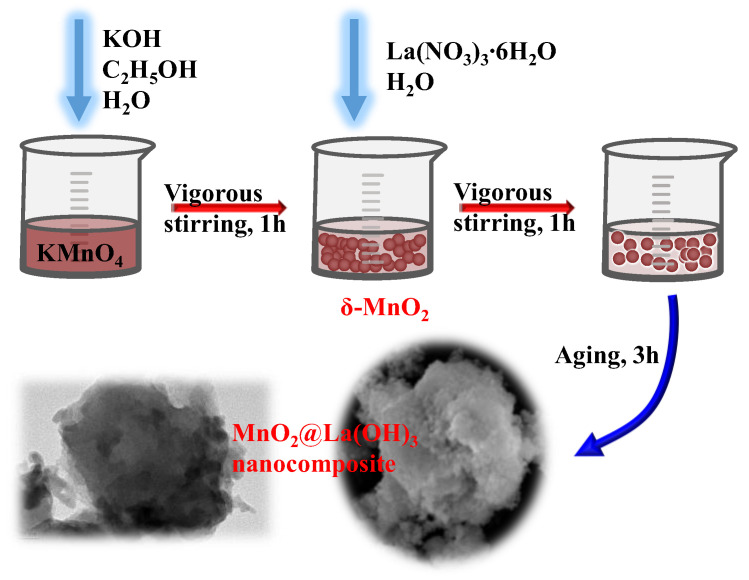
Schematic illustration of the preparation of core–shell structured MnO_2_@La(OH)_3_ nanocomposite.

**Figure 2 ijerph-19-10649-f002:**
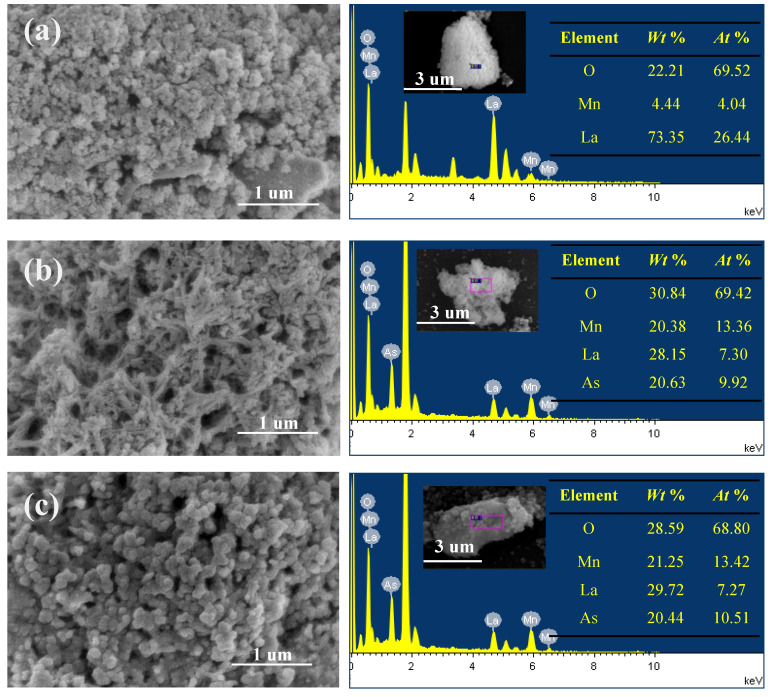
SEM micrographs and EDS analyses of core–shell structured MnO_2_@La(OH)_3_ nanocomposite before (**a**) and after the uptake of As(V) (**b**) and As(III) (**c**).

**Figure 3 ijerph-19-10649-f003:**
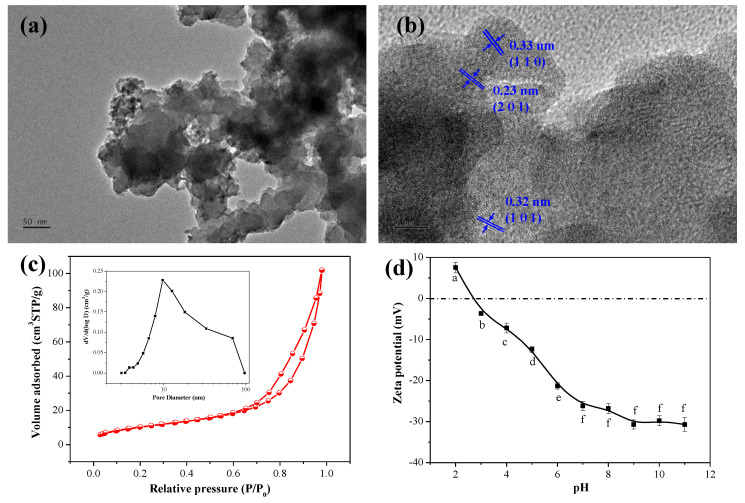
High-magnification TEM image (**a**), HR-TEM image (**b**), N_2_ adsorption–desorption analysis and pore width distribution in inset (**c**) and zeta potentials as a function of pH (**d**) of the core–shell structured MnO_2_@La(OH)_3_ nanocomposite. Different letters indicate significant differences in the ANOVA (α < 0.05).

**Figure 4 ijerph-19-10649-f004:**
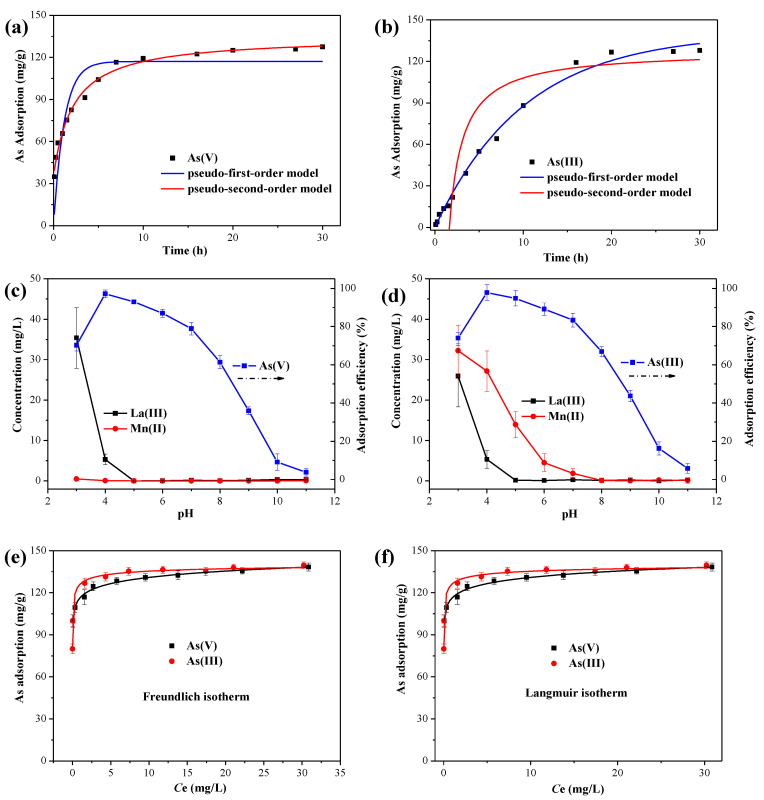
The influence of solution pH on As(V) (**a**) and As(III) (**b**) removal and the concentrations of La(III) and Mn(II) release at different pH conditions after As removal. The adsorption kinetics of As(V) (**c**) and As(III) (**d**) at pH 4.0. Adsorption isotherms at pH 4.0 with different initial As concentrations and their Freundlich (**e**) and Langmuir (**f**) fitting. Experimental conditions: the initial As concentration was 65 mg/L.

**Figure 5 ijerph-19-10649-f005:**
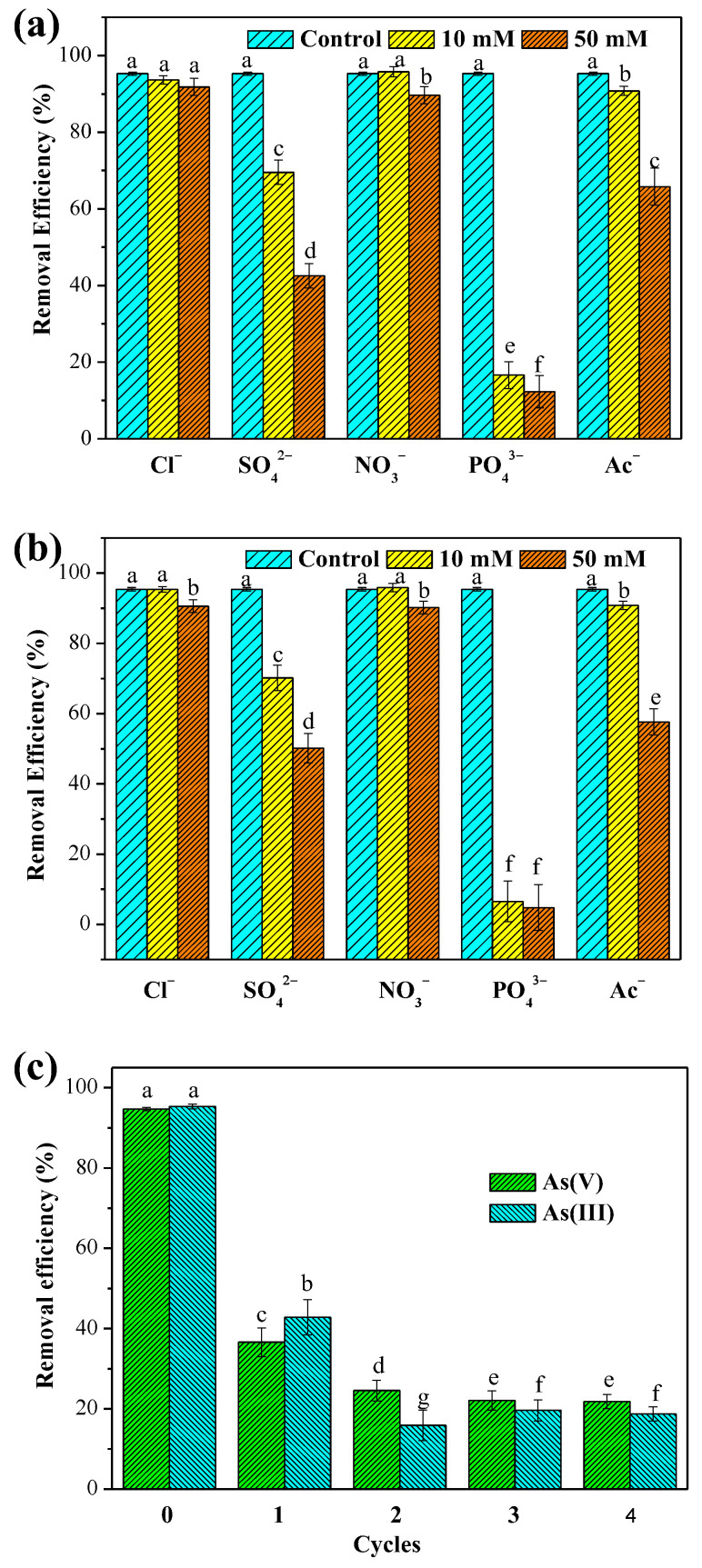
Effect of competitive anions on the removal of As(V) (**a**) and As(III) (**b**) at pH 4.0. (**c**) Reusability of MnO_2_@La(OH)_3_ nanocomposite on arsenic removal. Experimental conditions: the initial As concentration was 65 mg/L. The removal efficiencies of the control group were the removal efficiencies of As removal in the absence of the co-existing at pH 4.0. Different letters indicate significant differences in the ANOVA (α < 0.05).

**Figure 6 ijerph-19-10649-f006:**
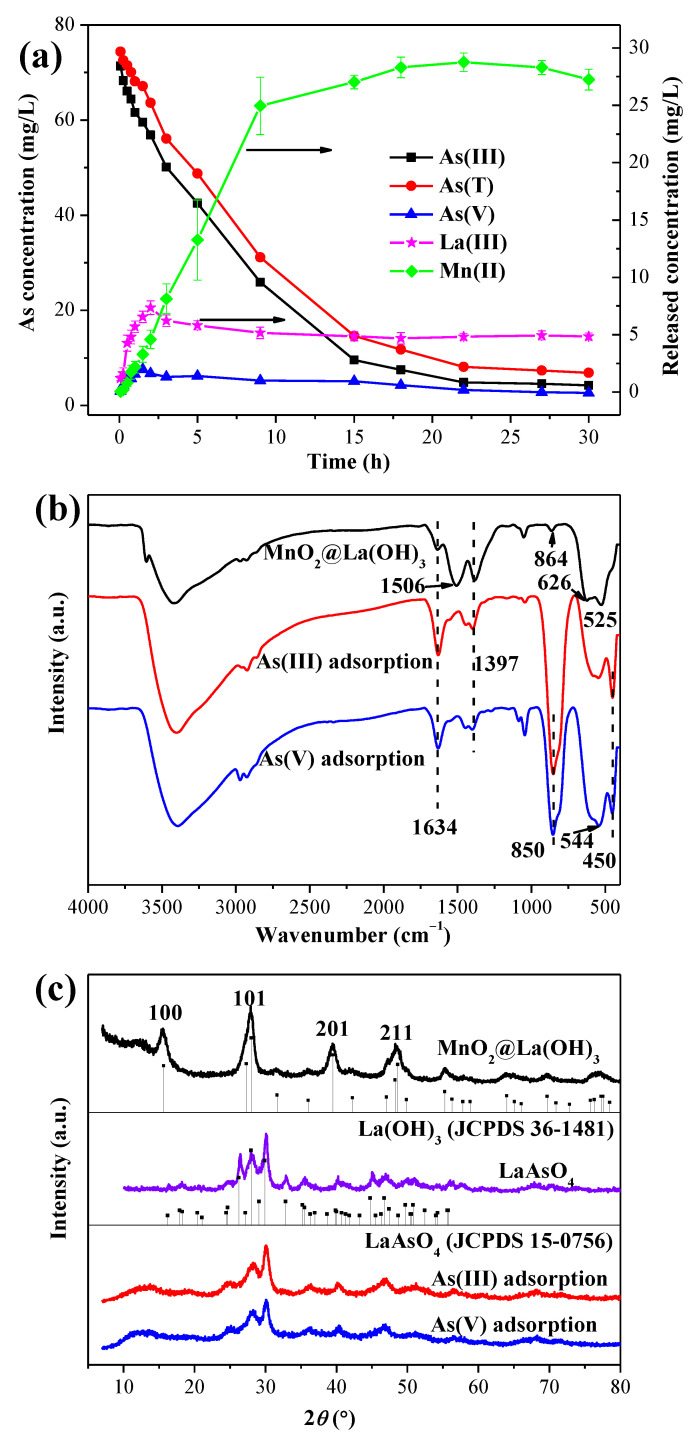
(**a**) Arsenite oxidation and adsorption kinetics by MnO_2_@La(OH)_3_ nanocomposite at pH 4.0. Experimental conditions: the initial As concentration was 75 mg/L. FTIR spectra (**b**) and XRD patterns (**c**) of the MnO_2_@La(OH)_3_ nanocomposite before and after As adsorption at pH 4.0.

**Figure 7 ijerph-19-10649-f007:**
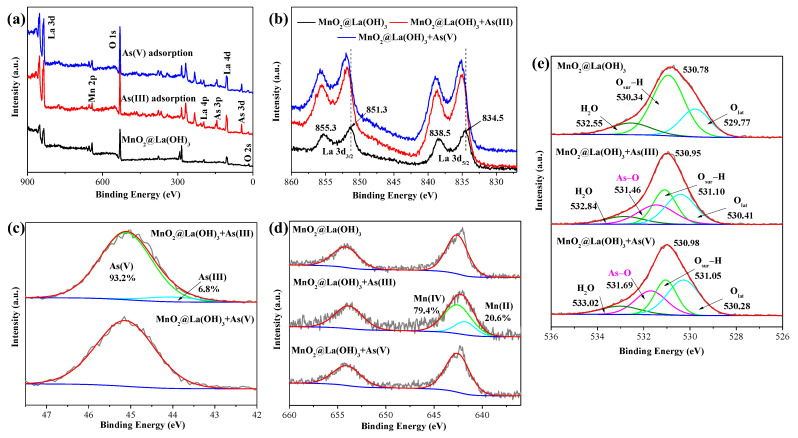
XPS analyses of MnO_2_@La(OH)_3_ nanocomposite before and after arsenic removal at pH 4.0. Full spectra (**a**), La 3d (**b**), As 3d (**c**), Mn 2p (**d**) and O 1s (**e**) high-resolution XPS spectra.

**Table 1 ijerph-19-10649-t001:** Adsorption kinetic parameters for As(V) and As(III) adsorption onto MnO_2_@La(OH)_3_ nanocomposite at pH 4.0.

Arsenic	Pseudo-First-Order Kinetic Model					Pseudo-Second-Order Kinetic Model				
	*k*_1_ (h^−1^)	*q*_e_ (mg/g)	*R* ^2^	MSE (mg/g)^2^	ARE (%)	*k*_2_ (g/(mg h))	*q*_e_ (mg/g)	*R* ^2^	MSE (mg/g)^2^	ARE (%)
As(V)	0.846	117.33	0.858	200.77	18.07	0.00872	129.87	0.995	100.83	12.52
As(III)	0.098	140.54	0.993	14.79	11.41	0.00946	127.82	0.892	892.61	262.22

**Table 2 ijerph-19-10649-t002:** Adsorption kinetic parameters for As(V) and As(III) adsorption onto MnO_2_@La(OH)_3_ nanocomposite at pH 4.0.

Arsenic	Langmuir Model					Freundlich Model				
	*k*_L_ (L/mg)	*q*_max_ (mg/g)	*R* ^2^	MSE (mg/g)^2^	ARE (%)	*k*_F_ ((mg/g)·(mg/L)^−1/*n*^)	1/*n*	*R* ^2^	MSE (mg/g)^2^	ARE (%)
As(V)	3.429	138.9	0.994	837.37	13.65	116.52	0.0521	0.990	1.51	0.80
As(III)	6.501	139.9	0.999	1330.3	33.75	126.23	0.0423	0.991	78.12	7.37

**Table 3 ijerph-19-10649-t003:** Comparison of the maximum adsorption capacities of As(V) and As(III) onto some adsorbents.

Adsorbent	pH	*q*_max_ (mg/g)		Ref.
		As(V)	As(III)	
MnO_2_@La(OH)_3_	4.0	138.9	139.9	This study
Mg-Fe-Ala-LDH	6.0	49.8	23.6	[8]
Ferrihydrite	3.0	142.86	n.a.	[12]
Fe–Mn composite	5.0	69.75	132.75	[41]
β-FeOOH NRs/CF monolith	6.0	172.9	103.4	[23]
CF@Mn-FeOOH	7.0	107.3	152.5	[31]
Fe–Mn composite oxide	7.0	31.68	59.44	[33]
α-FeOOH QDs@GO	n.a.	42.54	147.38	[24]
Mn–Fe binary oxide ^a^	–	50	50	[62]
Ca–Al–Fe ternary composites	n.a.	n.a.	56.86	[56]
Cerium oxide modified activated carbon	5.0	43.6	36.8	[55]
Ball-milled magnetite	n.a.	3.2	5.8	[63]
Cu–TiO_2_	7.5	19.719	24.244	[35]
FeMnOx/RGO	7.0	49.01	47.5	[64]
Hydrous cerium oxide modified graphene	4.0	62.33	n.a.	[65]

Note: n.a. means not available. ^a^ pH 4.4 for As(V) and 5.7 for As(III).

## Data Availability

The data presented in this study are available on request from the corresponding author.

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
