# Peer review of "Arsenic Oxidation and Removal from Water via Core–Shell MnO2@La(OH)3 Nanocomposite Adsorption"

_ijerph, 2022, doi:10.3390/ijerph191710649_

Round 1
Reviewer 1 Report
Dear editor and authors,
The manuscript entitled “Arsenic oxidation and removal from water via core–shell
MnO2@La(OH)3 nanocomposite adsorption”. assesses a topic that fits within the Journal. In general, I found the topic interesting and relevant to Int. J. Environ. Res. Public Health. However, after a deep revision of the manuscript, I found that some relevant information on the materials and methods is missing.
My main concern was associated with experimental design and statistical analysis of data. Hence, in my opinion, this manuscript is unpublishable in its current version, and it needs several improvements. Some reasons for this recommendation are explained below.
1.Introduction
Line 45. “to 10” instead “to10”
2. Materials and methods
Please, describe in detail the several phases of the study (Characterization, analysis of surface area, point zero charge, absorption kinetics) and which was the experimental unit, sample unit and measurement unit and statistical test to each phase.
Please describe in detail the models used to assess the absorption kinetics. Also, provide how the goodness of fit of used models was evaluated.
Regarding statistical analysis, the statistical model must be described in detail.
Results.
Line 220-221. The authors referred:
“As demonstrated in Fig. 3d, it can be clearly observed that the surface charge of the material significantly depended on the solution pH, and the pHpzc value was determined to be approximately 2.7.”
Which was the statistical test to determine that dependence between surface change and solution PH?
Line 270-290. The author compared the goodness of fit of the kinetic model only using the value of R2. However, R2 depicts several limitations when models with different mathematical structures are compared (mainly linear and non-linear models). For this reason, I recommended the use of other criteria to evaluate the goodness of fit as MSPE, AIC and BIC.
Line 398-400. The author referred:
“As presented in Figs. 5a and 5b, it can be obviously seen that the effects of co-existing anions on the removal efficiency of As(V) and 399As were similar to each other”
Please provide the statistical model and significance test to support this affirmation.
Figure 5. a,b,c. The control group was not defined in the m&m section. Also, this figure does not provide values of significance test.
Author Response
Response to Reviewer 1 Comments
The manuscript entitled “Arsenic oxidation and removal from water via core–shell MnO2@La(OH)3 nanocomposite adsorption”. assesses a topic that fits within the Journal. In general, I found the topic interesting and relevant to Int. J. Environ. Res. Public Health. However, after a deep revision of the manuscript, I found that some relevant information on the materials and methods is missing. My main concern was associated with experimental design and statistical analysis of data. Hence, in my opinion, this manuscript is unpublishable in its current version, and it needs several improvements. Some reasons for this recommendation are explained below.
Response 1: Thanks for your suggestion. We are truly grateful that the reviewer carefully examined our manuscript and provided constructive criticism. Thank you.
Point 1: Introduction
Line 45. “to 10” instead “to10”
Response 1: Thanks for the reminder. Sorry for our careless mistake. The word “to10” has been revised as “to 10”.
Point 2: Materials and methods
Please, describe in detail the several phases of the study (Characterization, analysis of surface area, point zero charge, absorption kinetics) and which was the experimental unit, sample unit and measurement unit and statistical test to each phase.
Response 2: Thanks for the valuable comments. The section “2.3. Characterization of adsorbents before and after As removal” has been added.
Please describe in detail the models used to assess the absorption kinetics. Also, provide how the goodness of fit of used models was evaluated.
Response 2: Thanks for your comments. The detail of the adsorption kinetics has been added in line 287. The coefficient of determination (R2), mean square error (MSE), and average relative error (ARE) were calculated to evaluate the fit quality.
Regarding statistical analysis, the statistical model must be described in detail.
Response 2: Thanks for your suggestion. The section “2.5. Statistical Analysis” has been added.
Point 3: Results.
Line 220-221. The authors referred:
“As demonstrated in Fig. 3d, it can be clearly observed that the surface charge of the material significantly depended on the solution pH, and the pHpzc value was determined to be approximately 2.7.” Which was the statistical test to determine that dependence between surface change and solution PH?
Response 3: Thanks for your comment. The surface change (zeta potential) as a function of pH was directly estimated by the Zetasizer Nano ZS. The average values of triple tests were used to determine the pHpzc value. In the manuscript, the detail of the determiation of the pHpzc value was provided in the section “2.3. Characterization of adsorbents before and after As removal”. The significance test has been added in the Fig. 3d.
Line 270-290. The author compared the goodness of fit of the kinetic model only using the value of R2. However, R2 depicts several limitations when models with different mathematical structures are compared (mainly linear and non-linear models). For this reason, I recommended the use of other criteria to evaluate the goodness of fit as MSPE, AIC and BIC.
Response 3: Thanks for the valuable advices. The coefficient of determination (R2), mean square error (MSE), and average relative error (ARE) were calculated to evaluate the fit quality.
Line 398-400. The author referred:
“As presented in Figs. 5a and 5b, it can be obviously seen that the effects of co-existing anions on the removal efficiency of As(V) and 399As were similar to each other”
Please provide the statistical model and significance test to support this affirmation.
Response 3: Thanks for your comments. An ANOVA with Tukey’s test was used and the significance test has been added in the Fig. 5.
Figure 5. a,b,c. The control group was not defined in the m&m section. Also, this figure does not provide values of significance test.
Response 3: Thanks for your comment. The removal efficiencies of As removal in the absence of the co-existing at pH 4.0 were used as the control group. The significance test has been added in the Fig. 5.

Reviewer 2 Report
General comments
This study focuses on the Arsenic oxidation and removal from water via core–shell MnO2@La(OH)3 nanocomposite adsorption. The topic is very interesting and it has practical importance. Though this topic is quite interesting and of practical significance, the manuscript requires revision in its several sections. The author must address the following issues to make it publishable.
Title
Title is fine and exactly reflects the insight of the study.
Abstract
The abstract is well structured. Can the author add its real life implications in brief?
Introduction
-Can you add a few latest related findings for the literature to show the research gap?
-There is no coherence in the description. Need to correct the coherence of the discussion.
-Need to avoid unrelated discussions.
-Need to address inconsistency in the text under the introduction section.
-Can you please add specific objectives at the last paragraph?
Materials and Methods
-Research approach is clear.
Results and Discussion
-The results seem correct. The author is strongly advised to findings of the related previous research and discuss superiority of the current findings.
Conclusion
Need to update this section by avoiding unnecessary text.
References
-Need to check the whole section and follow the journal style.
-Need to add recent related references in the concerned sections.
Author Response
Response to Reviewer 2 Comments
Point 1: General comments
This study focuses on the Arsenic oxidation and removal from water via core–shell MnO2@La(OH)3 nanocomposite adsorption. The topic is very interesting and it has practical importance. Though this topic is quite interesting and of practical significance, the manuscript requires revision in its several sections. The author must address the following issues to make it publishable.
Response 1: Thanks for your suggestion. We are truly grateful that the reviewer carefully examined our manuscript and provided constructive criticism. Thank you.
Point 2: Title
Title is fine and exactly reflects the insight of the study.
Response 2: Thanks for your comments. We are very much encouraged for your comments. Thank you.
Point 3: Abstract
The abstract is well structured. Can the author add its real life implications in brief?
Response 3: Thanks for your comments. We are very much encouraged for your comments. In the abstract, its real life implications such as As(III)-contaminated groundwater used for irrigation and As(V)-contaminated industrial wastewater have been added.
Point 4: Introduction
-Can you add a few latest related findings for the literature to show the research gap?
Response 4: Thanks for your advice. Some few latest related findings of the literatures have been added to show the advance of our research.
-There is no coherence in the description. Need to correct the coherence of the discussion.
Response 4: Thanks for your comment. The coherence of the discussion has been corrected.
-Need to avoid unrelated discussions.
Response 4: Thanks for your advice. Some unrelated discussions have been deleted.
-Need to address inconsistency in the text under the introduction section.
Response 4: Thanks for your comment. The inconsistency in the text under the introduction section has been revised.
-Can you please add specific objectives at the last paragraph?
Response 4: Thanks for your comment. The specific objectives have been added.
Point 5: Materials and Methods
-Research approach is clear.
Response 5: Thanks for your advice.
Point 6: Results and Discussion
-The results seem correct. The author is strongly advised to findings of the related previous research and discuss superiority of the current findings.
Response 6: Thanks for your comments. The superiority of the current findings compared with the related previous researches has been added.
Point 7: Conclusion
Need to update this section by avoiding unnecessary text.
Response 7: Thanks for your advice. Some unnecessary text has been removed.
Point 8: References
-Need to check the whole section and follow the journal style.
Response 8: Thanks for your advice. The whole references have been checked and some wrong references have been revised.
-Need to add recent related references in the concerned sections.
Response 8: Thanks for your comment. Some recent references have been added.

Round 2
Reviewer 1 Report
The manuscript entitled “Arsenic oxidation and removal from water via core–shell MnO2@La(OH)3 nanocomposite adsorption” has been improved, but some corrections and updates need to be made.
Materials and methods
L191. Please add the references of Freundlich and Langmuir models.
L192. The authors wrote: “The reference parameters were evaluated via nonlinear…”
In this process the parameters were fitted or estimated, the evaluation is the next step. I suggest changing the sentence as follows.
“The reference parameters were estimated via nonlinear…..”
L194. I suggest change “..to evaluate the goodness of fit” instead “..to evaluate the fit quality”
L194-195. I suggest rewrite the sentence as follow:
“The data were subjected to analysis of variance (ANOVA) using the General Linear Model and significance was declared with p<0.05. The post-hoc analysis was carried out using the Tukey's test.”
Results and discussion
L241. Remove the word “analysis” since the term ANOVA make reference to “analysis of variance”
L305-309. This information must be allocated in the materials and methods section
L402-407. This information must be allocated in the materials and methods section
L440-442. The different letters are defined by the post-hoc analysis (Tukey´s test). The ANOVA only support or reject the null hypothesis. We proposed the follow sentence:
“Different letters indicate significant differences (p < 0.05).”
The differences are between As(V) and As(III) or differences among cycles or both?
Author Response
The manuscript entitled “Arsenic oxidation and removal from water via core–shell MnO2@La(OH)3 nanocomposite adsorption” has been improved, but some corrections and updates need to be made.
Response: Thanks for your suggestion. We are truly grateful that the reviewer carefully examined our manuscript and provided constructive criticism. Thank you.
Point 1: Materials and methods
L191. Please add the references of Freundlich and Langmuir models.
Response 1: Thanks for the advice. The references of Freundlich and Langmuir models have been added.
L192. The authors wrote: “The reference parameters were evaluated via nonlinear…”
In this process the parameters were fitted or estimated, the evaluation is the next step. I suggest changing the sentence as follows.
“The reference parameters were estimated via nonlinear…..”
Response 1: Thanks for the comment. The word “evaluated” in the sentence has been changed as “estimated”.
L194. I suggest change “..to evaluate the goodness of fit” instead “..to evaluate the fit quality”
Response 1: Thanks for the valuable comment. The words “the goodness of fit” have instead “the fit quality”.
L194-195. I suggest rewrite the sentence as follow: “The data were subjected to analysis of variance (ANOVA) using the General Linear Model and significance was declared with p<0.05. The post-hoc analysis was carried out using the Tukey's test.”
Response 1: Thanks for the valuable advice. The sentence has been rewritten as mentioned above.
Point 2: Results and discussion
L241. Remove the word “analysis” since the term ANOVA make reference to “analysis of variance”
Response 2: Thanks for the valuable comments. The word “analysis” has been removed.
L305-309. This information must be allocated in the materials and methods section
Response 2: Thanks for your comments. This information has been allocated in the section “Statistical Analysis”.
L402-407. This information must be allocated in the materials and methods section
Response 2: Thanks for your suggestion. This information has been allocated in the section “Statistical Analysis”.
L440-442. The different letters are defined by the post-hoc analysis (Tukey´s test). The ANOVA only support or reject the null hypothesis. We proposed the follow sentence:
“Different letters indicate significant differences (p < 0.05).”
The differences are between As(V) and As(III) or differences among cycles or both?
Response 2: Thanks for the valuable comment. The differences are both between As(V) and As(III) and among cycles. We are quite sorry for some mistakes due to our careless and knowledge.
